# Mapping the Transcriptional and Fitness Landscapes of a Pathogenic *E. coli* Strain: The Effects of Organic Acid Stress under Aerobic and Anaerobic Conditions

**DOI:** 10.3390/genes12010053

**Published:** 2020-12-31

**Authors:** Francesca Bushell, John M. J. Herbert, Thippeswamy H. Sannasiddappa, Daniel Warren, A. Keith Turner, Francesco Falciani, Peter A. Lund

**Affiliations:** 1School of Biosciences and Institute of Microbiology and Infection, University of Birmingham, Birmingham B15 2TT, UK; cesca.bushell@gmail.com (F.B.); drthippeshhs@gmail.com (T.H.S.); 2Institute of Systems, Molecular and Integrative Biology, University of Liverpool, Liverpool L69 3BX, UK; herbertjmj@gmail.com (J.M.J.H.); sgdwarre@student.liverpool.ac.uk (D.W.); 3Quadram Institute Biosciences, Rosalind Franklin Road, Norwich Research Park, Norwich NR4 7UQ, UK; keith.turner@quadram.ac.uk

**Keywords:** acid stress, organic acid, SCFA, short-chain fatty acid, gastrointestinal tract, uropathogenic *E. coli*, RNAseq, TraDIS

## Abstract

Several methods are available to probe cellular responses to external stresses at the whole genome level. RNAseq can be used to measure changes in expression of all genes following exposure to stress, but gives no information about the contribution of these genes to an organism’s ability to survive the stress. The relative contribution of each non-essential gene in the genome to the fitness of the organism under stress can be obtained using methods that use sequencing to estimate the frequencies of members of a dense transposon library grown under different conditions, for example by transposon-directed insertion sequencing (TraDIS). These two methods thus probe different aspects of the underlying biology of the organism. We were interested to determine the extent to which the data from these two methods converge on related genes and pathways. To do this, we looked at a combination of biologically meaningful stresses. The human gut contains different organic short-chain fatty acids (SCFAs) produced by fermentation of carbon compounds, and *Escherichia coli* is exposed to these in its passage through the gut. Their effect is likely to depend on both the ambient pH and the level of oxygen present. We, therefore, generated RNAseq and TraDIS data on a uropathogenic *E. coli* strain grown at either pH 7 or pH 5.5 in the presence or absence of three SCFAs (acetic, propionic and butyric), either aerobically or anaerobically. Our analysis identifies both known and novel pathways as being likely to be important under these conditions. There is no simple correlation between gene expression and fitness, but we found a significant overlap in KEGG pathways that are predicted to be enriched following analysis of the data from the two methods, and the majority of these showed a fitness signature that would be predicted from the gene expression data, assuming expression to be adaptive. Genes which are not in the *E. coli* core genome were found to be particularly likely to show a positive correlation between level of expression and contribution to fitness.

## 1. Introduction

The ability to predict the phenotype of an organism from knowledge of its genotype is one of the most desirable, and one of the hardest to achieve, targets in genetics. In theory, a complete deterministic model of the genome including the properties of every entity it encodes, and how they interact with each other and with every aspect of their environment would enable the prediction of the behaviour and properties of any organism in any environment from a knowledge of its genome alone. In practice, such models are far out of reach, and although remarkable progress has been achieved in developing methods for inferring phenotypes from large-scale datasets [1,2,3,4,5], significant obstacles remain to the development of improved predictive models of organisms based on knowledge of their genome sequences. One of these obstacles is that, even for the best-characterised and simplest organisms, the roles of many of their genes are still obscure or completely unknown [6,7]. Indeed, even the prediction of which open reading frames in a genome actually encode entities with biological function such as proteins or regulatory RNAs is still not completely accurate [8]. A second obstacle is that mechanistic models need parameters, but accurate parameterisation is notoriously difficult [9]. A third obstacle is that, even though omics technologies enable many different properties at different levels to be measured simultaneously with varying degrees of accuracy, it is often not clear how these measurements relate to the actual contribution of individual genes to fitness, a fact that is demonstrated by the surprisingly poor correlations that are often found between these measurements [10,11,12]. We still do not fully understand the way that genetic information is regulated and integrated to produce a functioning organism. Much hope has been put in the application of ML/AI models that bypass to some extent the needs to understand the mechanisms in full and which may provide prediction and hypothesis at the same time.

A direct way to estimate the contribution of a gene to the overall phenotype of an organism under a particular condition is to determine how loss of the gene affects this phenotype: this is effectively the basis of most genetic analysis. The impact can be measured by competing the mutant against the wild type, thus directly measuring their relative fitness in any chosen condition [13]. Such information is empirical rather than mechanistic: it tells us how important a gene may be under a particular condition, but not about the reasons for this importance. This is, nonetheless, very useful in model building, particularly if data can be obtained for multiple genes, as they enable genes to be clustered according to their impact on specific phenotypes. Relative fitness data for large numbers of genes have, however, often been laborious to obtain, being derived initially from studies on single mutants (ideally deletions), and more recently from experiments involving screening libraries of single-gene knockouts, transposon insertions, or barcode-tagged genes. These can either be studied individually (usually using automation), or by pooling and screening before and after exposure to a given condition, to identify genes which are important for growth under that condition [14,15,16]. The latter approach has recently become much more powerful through the development of methods that involve analysis of very high density transposon libraries using next-generation sequencing methods. This approach, usually called Tn-seq or TraDIS, can give a detailed insight into the relative contribution of all non-essential genes to fitness under a particular condition, as well as defining which genes are essential or non-essential in the first place [17,18]. Such methods entail the simultaneous competition of a very large number of mutants against each other, with scoring of their relative numbers being determined from the frequency of sequencing reads corresponding to each mutant. This approach benefits both from the high capacity of massively parallel sequencing methods, and also from the fact that information about each gene is derived from the averaged data of many different transposon insertions in them, rather than from a single knockout.

An important question then arises which is how well do fitness measurements obtained from these high-throughput methods correlate with the data obtained from other methods: in other words, to what extent do omics approaches give useful information about contributions of genes to an organism’s fitness? We wished to address this question for conditions that resembled one that the organism in question has been exposed to and under which it has hence been selected for efficient growth, and the current paper reports the initial result of such an analysis. For the two methods, we chose TraDIS to provide fitness data, and transcriptomics (RNAseq) to provide expression data. The rationale for making this choice was that other omics methods, although they measure properties which are arguably closer to those directly relevant to organismal function, are not as comprehensive in their measurements, whereas RNAseq captures gene expression information over a wide dynamic range. The organism we chose was a pathogenic strain of *Escherichia coli* (EO499 of the ST131 clade, also known as NCTC 13441). This is a multiple antibiotic resistant uropathogenic (UPEC) strain of *E. coli* that can cause a range of infections, including but not limited to urinary tract infections [19]. It is known that the human gut can act as a reservoir of ST131 and other UPECs for recurrent UTIs; therefore, a study of how a member of this group responds to some conditions found in the gut may give insights that could help in eliminating these strains from the gut microbiome of individuals suffering from such infections [20,21]. We exposed this strain to different combinations of stresses related to those found in the gut, and the current study reports the results from a subset of these. Specifically in this study, we looked at combinations of three parameters: pH, the presence or absence of different short-chain fatty acids SCFAs (acetic, propionic, and butyric), and either aerobic or anaerobic conditions.

All of these are related to conditions encountered during the passage of this particular bacterium through the human gut, which is part of its natural environment. *E. coli* has to cope with a wide range of acidic challenges as it passes through the human gut, with a very low pH in the stomach caused by secretion of HCl, plus SCFAs in parts of the intestine arising from bacterial metabolism of gut contents. Acid washes are also used in some food preparation pipelines where *E. coli* contamination may be a risk [22]. For this reason, both its ability to survive different pH conditions (its intrinsic acid resistance) and its responses to low pH (its inducible acid tolerance) have been extensively studied (for reviews, see [23,24,25,26]. Different isolates of *E. coli* show high variability in acid resistance, which does not correlate simply with pathogenicity [27,28,29], and even in the same strain the acid resistance is dependent on what is present in the growth medium, and the stage of growth, with stationary phase cells generally being more resistant than those in the exponential phase [24]. Much attention has focused on the four main inducible response systems (called AR1–AR4) that when induced give *E. coli* the ability to survive better under low pH, often by importing an amino acid, decarboxylating it, and exporting the product. The effects of SCFAs have also been studied and it is well established that they become significantly more inhibitory to bacterial growth as the pH of the solution drops, in part due to their ability to cross membranes when in their undissociated, uncharged form [30,31]. Most studies on the effects of acids on *E. coli* have been performed under aerobic conditions, and there is evidence that responses to low pH may be different in many ways under anaerobic conditions [32,33,34].

Our aim in this study was thus: first, to investigate the transcriptional changes that occur in response to different combinations of pH, SCFA, and aeration; second, to see which genes and pathways are important for fitness under these same conditions; and third, to compare these datasets and to deduce which component of the transcriptional response to stress is adaptive and, therefore, has probably evolved to increase fitness and which part may be part of an adverse response to stress exposure. Both the methods used generate large datasets, and our aim in this paper is to provide an overview analysis of these rather than to scrutinise particular genes or pathways in detail.

## 2. Materials and Methods

### 2.1. Growth Conditions

*E. coli* serotype ST131 strain EO499 [35] was grown in M9 minimal medium supplemented with 0.4% (*w*/*v*) glucose and 0.2% (*w*/*v*) cas-amino acids (Fisher Scientific) (referred to from hereon as ‘M9supp’). All growth media were buffered with 100 mM MOPS and 100 mM MES as described in [36]. M9 salts (starting pH 5.8) were brought to the relevant pH using either hydrochloric acid or sodium hydroxide added drop by drop until the desired pH was reached. The salts were then autoclaved and supplements were added subsequently; this did not cause any change in pH.

All experiments (measurement of growth, and growth of cultures for RNAseq or TraDIS) were performed in triplicate for each condition. Three 5 mL aliquots of M9supp at pH 7 were initially each inoculated from a different single colony and grown overnight at 180 rpm, at 37 °C in 25 mL universal tubes. Three 250 mL Erlenmeyer flasks containing 50 mL M9supp at pH 7 were pre-warmed and each was inoculated from one overnight culture to a starting OD_600_ of 0.05. Flasks were shaken at 180 rpm, at 37 °C. Anaerobic cultures were grown at 37 °C in a Don Whitley Scientific anaerobic cabinet with a gas mix of 5% CO2/95% N2. In total, data were gathered for sixteen different combinations of conditions (the variables being aerobic or anaerobic, pH 7 or pH 5.5, and without added organic acid or with one of acetic, propionic or butyric acid added).

### 2.2. Construction of TraDIS Library

The construction of the TraDIS library in the ST131 clone EO499, using a mini-Tn5 transposon encoding chloramphenicol resistance, has been described [37]. The library has approximately 450,000 unique insertion sites across the genome (5.4 Mba), an average density of 1 insertion every twelve bases.

### 2.3. RNA Extraction, Sequencing, and Data Analysis

Growth was monitored spectroscopically until an OD_600_ of 1.8 was reached. A dilution and replenishment protocol was then applied to achieve steady state (balanced) growth as described previously [38], by removing 5 mL of culture and replacing it with 5 mL of pre-warmed M9supp. This was performed every 30 min for one hour for the anaerobic cultures, and every fifteen minutes for the aerobic cultures; pilot experiments established that this maintained the culture density between an OD600 of 1.8 and 2. To impose the stress, 5 mL of culture was then removed, and 5 mL of pre-warmed M9supp at the appropriate pH with or without additional SCFA was added to create the specific stress condition (thus the stress was applied to the 50 mL cultures). Growth was then continued for 15 min. A volume of 5 mL of culture was then removed from each flask and added to an equal volume of Qiagen RNA Protect and mixed thoroughly. Cells were pelleted by spinning at 4000 rpm for 5 min in a chilled centrifuge, and pellets were stored at −80 °C for downstream RNA purification.

RNA was purified from the pellets collected as above using Qiagen RNeasy kits according to the manufacturer’s instructions. The optional on-column DNA digestion step was performed for all samples to ensure there was no DNA contamination of samples. PCR amplification of the *evgS* gene was used to confirm the absence of genomic DNA. All samples were checked for quality using a Prokaryote Total RNA Nanochip, and analysed using the Agilent Bioanalyser in order to provide a RNA integrity number (RIN). Only samples with a RIN of 7 or higher were used for sequencing. Ribo-depletion was performed using the Illumina Ribo-zero rRNA removal kit according to the manufacturer’s instructions. All samples of a satisfactory standard and final yield were then transferred to the Centre for Genomic Research at the University of Liverpool for sequencing, which was performed using the Illumina 2500 HiSeq platform.

To reduce sequencing costs, the cultures grown at either pH 7 or pH 5.5 were sequenced individually to produce three biological replicates per condition. Then, for each of the SCFAs, three biological replicates were grown in separate flasks, after which the RNA was extracted and pooled prior to single-stranded RNAseq paired-end sequencing, with each pool containing equal concentrations of RNA from each replicate. R1 and orphaned reads from the paired-end sequencing data were combined and mapped to a Prokka-annotated EO499 genome sequence using STAR v2.5.3a_modified [39], and mapped reads were assigned to *E. coli* features using featureCounts v1.5.2 [40]. RUVSeq was then used in conjunction with ERCC B spike-in controls to remove any batch effects arising from different sequencing preparations and runs of the biological replicate samples, where K was set to 2 [41]. Then, to find significantly differentially expressed genes between any pools of SCFAs and matched pool controls, the Voom function from the Limma Bioconductor package was employed [42]. Briefly, each organic pool contrast analysis contained 8 samples and 4 treatment groups (the combined 6 biological replicates of pH 7 and pH 5.5 samples, plus the two pooled samples to be compared). A constant read count of 1 was added to the 8 samples prior to Gene-wise trimmed median of means (TMM) normalization. Then a Limma contrast, a paired moderated t-test, was performed between the pools using the information from the pH 7 and pH 5.5 replicated treatment groups to calculate statistical significance between the pools. Log2-fold changes from the RNAseq pipeline were used as signatures for subsequent GSEA analyses, looking for KEGG pathway enrichments.

### 2.4. TraDIS Methods

To grow cultures for TraDIS analysis, 10 µL of the EO499 transposon mutant library was added to 50 mL M9supp pre-adjusted to the required pH (7 or 5.5) and containing, where necessary, the relevant SCFA at the appropriate concentration, giving a starting OD_600_ of 0.01. Cultures were grown aerobically or anaerobically as described above for 12 h. If the OD_600_ had reached at least 1 by this point, 10 mL of culture was removed from the sample, pelleted, and stored as for the RNAseq experiments. If the OD_600_ was lower than 1 after 12 h, growth was continued for a further 12 h and another sample of 10 mL was centrifuged and frozen as described. Three biological replicates were performed for each combination of conditions. Genomic DNA was then purified using the Qiagen DNeasy Blood and Tissue kit according to the manufacturer’s instructions. Preparation of sequencing libraries and sequencing was performed as described [43].

For analysis of TraDIS data, we used ESSENTIALS [44] to calculate the statistical significance of relative fitness values for all genes between stress and control conditions. Genes which show a decreased transposon read count after growth in a given stress condition relative to a control are assumed to make a greater contribution to strain fitness under the stress than under control conditions, and vice versa. We have avoided use of the common term “conditionally essential” to describe these genes, as the loss of function of these genes that is assumed to occur when transposons are inserted in them does not always lead to a complete failure to grow under particular stress conditions, which would be the usual definition of essentiality, but just to a slower rate of growth. Instead, we refer to “relative fitness”. All relative fitness levels calculated from TraDIS scores are expressed as control/stress, as we are making comparisons with RNAseq data where it is generally assumed that the higher the relative expression (stress/control), the more important the gene may be. This means that a high positive relative TraDIS score in our calculations represents a strong contribution of the gene to fitness under the stress condition (hence, strains carrying transposons in these genes will be expected to be depleted after growth under the stress relative to the control). The full results of the analysis for all conditions with both RNAseq and TraDIS are in Appendix A. All data have been deposited with the European Bioinformatics Institute (ArrayExpress) with the accession number E-MTAB-9762.

The overall structure of the investigation is summarised in Appendix A. 

### 2.5. Additional Bioinformatics Analysis

For both the RNAseq and TraDIS data, the Euclidean distance between the Spearman correlated log-fold ratios were used to determine the strength of association among genes and treatment comparisons. Based on this distance metric, hierarchical clustering was achieved through the agglomerative unweighted pair group method with arithmetic mean (UPGMA). Following this, principal component analysis was carried out to further explore differences observed between treatment responses. This analysis was performed using the stats package of R [45].

To assess KEGG pathway activation or repression under the different stresses, leading edge gene set enrichment analysis [46] was performed using gene sets consisting of *E. coli* MG1655 KEGG pathway genes, which were enriched on to 16 log2 fold change differential gene expression stress signatures and 16 TraDIS stress signatures. The Broad Institute command line tool was used and the output was parsed to produce a 2D clustered heatmap as presented in the Appendix A. GSEA results were hierarchically clustered using Euclidean distance and the average agglomeration clustering method. Bar plots and 2D clustering plots were made in R using the heatmap.2 from the gplots R package and ggplot function from the ggplot2 R package [47,48].

## 3. Results

### 3.1. Determination of Stress Conditions

Four considerations determined the choice of stress levels used in this study. First, we wanted the stressors used to be at concentrations close to those that exist in the human gut, so as to produce data that are biologically relevant. Second, for TraDIS to be possible, bacteria generally have to go through multiple generations, in order for the selective effects of different mutations to be detectable; therefore, significant growth was required after imposition of the stress. Third, both RNAseq and TraDIS required us to be able to harvest enough cells for extraction of a sufficient amount of material for subsequent processing. Finally, we wanted the impact of the different stresses on growth rate to be approximately the same, so that the data were not significantly confounded by the consequences of large variations in growth rate, which can affect variables such as RpoS levels and hence could itself lead to apparent differences in stress responses that are not related to the actual stressors [49].

We, therefore, performed a series of preliminary growth experiments, where EO499 was grown as described in Materials and Methods at either pH 7 or pH 5.5 with or without SCFAs at a range of different concentrations. Examples of typical growth curves can be seen in Appendix A. As expected, we found that SCFAs reduced growth more when the pH was more acidic; this is a well-known effect and related to the greater degree of undissociated and hence membrane-permeable SCFAs present as the pH approaches the pKa of the acid [30,31]. The effects of the different SCFAs on growth was in the order butyric > propionic > acetic at both pH 7 and 5.5. Gut concentrations of these acids vary between individuals and at different positions within the gut, but are generally in the order acetic > propionic > butyric [50,51]. The final conditions chosen were: at pH 7, 40 mM acetic acid, 30 mM propionic acid, or 30 mM butyric acid, and at pH 5.5, 4 mM acetic acid, 2 mM propionic acid, or 2 mM butyric acid. These are quite close to those used in a previous study that aimed to replicate the relative concentrations of SCFAs in the gut [27]. The same concentrations were used whether cells were grown aerobically or anaerobically. The pH of the cultures was monitored at the end of all growth experiments, and did not drop more than a maximum of 0.3 units below the starting pH in any case.

### 3.2. Analysis of Gene Expression

We used RNAseq to analyse short-term changes in relative gene expression as affected by two variables: the pH, and the nature of the SCFA (if one was included). A fifteen minute time point was used, as our previous time course study which was performed under similar conditions had shown that the changes in gene expression were highest at this point [38]. We collected data from cells which had been grown either aerobically or anaerobically for at least four hours before data collection. Because of the large number of different variables studied, we generated the RNAseq data by repeating each growth experiment in triplicate, preparing the RNA from each experiment, and then pooling it before sequencing and statistical analysis as described in Materials and Methods. The complete results of the analysis of RNAseq data and TraDIS data for all conditions can be found in Appendix A.

We found that the effect of pH change alone on gene expression was small under these conditions. When grown aerobically, significant up-regulation or down-regulation was seen for only twelve and seven genes, respectively, when treated for fifteen minutes at pH 5.5 relative to expression at pH 7 (significance in this case was taken as a log_2_fold change of >1.5 or <−1.5 and an adjusted *p* value of 0.05 or less; Appendix A), and no significant changes of expression for any genes were seen under anaerobic conditions. Relaxing the stringency of the cut off to include all genes showing any increase in expression at an adjusted *P* value of 0.05 or less would add only two more genes to this list: *glpD* and *yhhA* (with log_2_fold changes 1.28 and 1.16, respectively). Previous studies of low pH-induced changes in gene expression have often seen much larger numbers of gene changes than this, and possible reasons for the discrepancy with the current study are considered in the Discussion. It was notable that most of the up-regulated genes were those associated with stress responses, including the AR4 acid shock response that converts imported lysine to cadaverine; however, none of the genes of the AR2 or AR3 systems showed significant up- or down-regulation. The significantly repressed genes are largely involved with arginine uptake or biosynthesis which is intriguing given the role of arginine in the operation of the AR3 acid resistance system and the reported importance of arginine biosynthetic genes in UPEC pathogenesis, but the reason for their repression under these conditions, which does not appear to have been reported previously, are not clear [24,25,26,52,53].

To investigate the interaction of pH and aerobic status with individual SCFAs on gene expression of EO499, we determined the relative fold change in expression for all genes in the presence or absence of each SCFA at pH 7 or pH 5.5 under both aerobic and anaerobic conditions. Unlike pH alone, exposure to SCFAs in all tested conditions was a powerful transcriptional modulator. In order to determine whether there were general trends common to all SCFAs in the different conditions (pH and oxygen), we converted these data into simple ratios (expression in presence SCFA/expression in absence of SCFA for each condition) then clustered the data as described in Materials and Methods with the result shown in Figure 1. All genes except for rRNAs and small RNAs were used in this cluster analysis irrespective of the adjusted p-value of the expression change, but unlike the case with low pH alone, large numbers of highly significant changes in gene expression were seen in these conditions. We also included in this cluster analysis ratios of expression levels for all genes compared between anaerobic and aerobic cultures at either pH 7 or pH 5.5, in the absence of any SCFA. In addition, we performed a principle component analysis of the relative changes in gene expression shown by the RNAseq data under all conditions, with the results shown in Figure 2.

These analyses show that the most important factor in determining gene expression changes was the aeration state of the culture. Thus, in the presence of the different SCFAs but irrespective of the SCFA used or the pH of the culture, all samples from aerobic conditions clustered together and all from anaerobic conditions clustered together. The different SCFAs cluster together in both cases, indicative of a conserved response to the stress imposed by these acids irrespective of their nature. The sole exception to this was the impact of propionic acid on anaerobic cultures at pH 5.5 where there was relatively little change in gene expression. The PCA analysis also shows clear separation of the aerobic and anaerobic cultures in the presence of SCFAs on the x axis; as expected from the cluster analysis, propionic acid under anaerobic conditions at pH 5.5 is an outlier with greater separation from the other SCFAs in the second component. Both analyses also show strikingly similar gene expression changes between cultures grown anaerobically in the absence of any SCFA with those seen in cultures where SCFAs were added under aerobic conditions: in other words, the gene expression changes taking place from the short-term stress caused by SCFAs added under aerobic conditions are very similar to the expression changes that occur when cells are grown anaerobically long term, irrespective of pH. This point is specifically illustrated in Appendix A, which shows how changes in gene expression are highly correlated between acetic acid addition to aerobic cultures, and anaerobic growth, at both pH values. Similar results were seen for the other SCFAs. The pH of the culture had relatively little effect on gene expression in the presence of the different SCFAs under aerobic conditions, although there is some separation in the second component of the PCA analysis.

Both cluster and PCA analysis show a very clear separation of gene expression caused by addition of SCFAs that is dependent on whether cultures were growing aerobically or anaerobically. Strikingly, the cluster analysis shows that many expression changes were reversed when SCFAs were added under anaerobic conditions, although the extent of the changes was less under anaerobic conditions: genes that were induced by SCFA addition under aerobic conditions were often repressed under anaerobic conditions, and vice versa.

To investigate the nature of the changes in more detail, we used gene set enrichment analysis (GSEA) of KEGG pathways to determine the pathways that were most significantly up- or down-regulated under the different conditions of growth based on their transcriptional signatures. The full results of this analysis are tabulated in Appendix A. We then clustered the most significantly enriched pathways of the pathways that were enriched in at least two conditions. The result of one of these analyses is shown in Figure 3 for illustration; all four of the analyses (pH 7 and pH 5.5, aerobic and anaerobic) are shown in Appendix A.

As expected from the cluster and PCA analysis of the individual genes, there was a substantial overlap in significant pathways detected by GSEA under all the cases where SCFAs were added to aerobic cultures, irrespective of pH (compare Appendix A). Overlap was also seen where SCFAs were added to anaerobic cultures, although in these cases fewer pathways overall were detected as showing significant change (compare Appendix A (aerobic) with S3C,D (anaerobic)). For example, under aerobic conditions plus SCFAs, irrespective of pH, synthesis of ribosomal proteins and of amino-acyl tRNAs were both significantly down-regulated, suggestive of a general suppression of translation. Likewise, pathways for a range of features associated with aerobic respiration (oxidative phosphorylation, sulphur relay system, quinone biosynthesis, and thiamine metabolism) were down-regulated by addition of SCFAs to aerobic cultures, as were general pathways such as homologous recombination, mismatch repair, and peptidoglycan biosynthesis. All these pathways were also down-regulated in the anaerobic cultures compared to the aerobic cultures at both pH values. Pathways that were likewise generally up-regulated with the addition of SCFAs to aerobic cultures and also in the anaerobic cultures indicated a broad metabolic switch in these cultures: several pathways for sugar metabolism and uptake, amino acid degradation, and alternative carbon metabolism were up-regulated, as were the genes for flagellar assembly.

There were fewer pathways in common between the transcriptional responses to the different SCFAs in anaerobic cultures, as expected from the smaller number of gene expression changes that occurred in these, in particular the low level of changes in expression that occurred in the presence of propionic acid grown anaerobically at pH 5.5. We looked in particular for pathways that showed evidence of switched regulation between the aerobic and anaerobic responses to SCFAs, given the reversal in transcription responses that was apparent from the cluster analysis shown in Figure 1. Several pathways showed evidence of such behaviour: DNA replication, homologous recombination, mismatch repair, protein export, purine metabolism, ribosomal synthesis, and the sulphur relay system were all down-regulated by the addition of SCFAs to aerobic cultures, and also in anaerobic cultures in the absence of SCFAs, but up-regulated in many of the SCFA anaerobic cultures, in all cases relative to the controls. Conversely, flagellar assembly, galactose metabolism, the PTS sugar transport system, and pentose metabolism, were relatively up-regulated in the majority of the SCFA aerobic cultures and in the anaerobic cultures, but down-regulated in at least some of the SCFA anaerobic cultures. Potential biological interpretations of all these results will be considered further in the Discussion.

A final point to note with transcription was that a significant number of changes occurred in genes that are not part of the *E. coli* core genome but which are annotated here as ST131 specific genes. We cannot analyse the biological consequences of this in detail as in most cases their functions are unknown, but Appendix A shows that over a hundred of these genes were up-regulated either by SCFAs in aerobic cultures, or by anaerobic growth, irrespective of pH, and there were significant overlaps between the different conditions. This implies a significant level of transcriptional response to stress that may be specific to this particular group of *E. coli* strains, though it is impossible to generalise without more data, and these genes may repay further investigation to determine whether any are important for adaptation to growth in the human gut. We repeated the cluster analysis with these genes, and the result (shown in Appendix A) shows a very similar pattern to that seen with cluster analysis of all genes. This observation raised the possibility that a similar pattern might be seen when contributions to gene fitness were analysed, and this point is considered further below in Section 3.5.

### 3.3. Analysis of Contributions of Genes and Pathways to Strain Fitness

For analysis of the TraDIS data, we adopted essentially the same approach that we used with the RNAseq data. Cultures were processed for TraDIS analysis following prolonged growth as described in Materials and Methods under the specific stress condition. Following analysis of the raw data using ESSENTIALS, TraDIS results were expressed as relative fold change, using values of control/stress as explained above so that genes where transposon inserts generally cause a loss of fitness under a specific conditions, and so are assumed to be important for growth in that condition, have a positive value. Data were clustered and analysed using PCA in the same way as for RNAseq, with the results shown in Figure 4 and Figure 5. The results are more complex than seen with the RNAseq. Nonetheless, some clear patterns emerge.

From the cluster analysis shown in Figure 4, at the top level, it can be seen that the results from all anaerobic cultures treated with SCFA cluster separately from all other conditions irrespective of pH, and furthermore that these data form two clusters depending on the pH of the culture. At the next level, five clear groups of related relative fitness profiles can be seen. These are (from left to right across Figure 4): SCFAs under aerobic conditions at pH 7 plus two conditions with no added SCFA (pH 5.5 vs. pH 7 aerobic, and pH 7 aerobic vs. anaerobic; cluster I); two conditions lacking SCFAs (pH 5.5 vs. pH 7 anaerobic, and pH 5 anaerobic vs. aerobic; cluster II); added SCFAs under aerobic conditions at pH 5.5 (cluster III); and then the two clusters referred to above: added SCFAs under anaerobic conditions at pH 5.5 (cluster IV), and added SCFAs under anaerobic conditions at pH 7 (cluster V). Broadly the same result is seen with the PCA analysis, with clear separation between the two different pH values for the anaerobic cultures with added SCFA, and between all anaerobic and all aerobic cultures with added SCFA. Overall, although the data are less clearly demarcated than was the case with the RNAseq analysis, the relative fitness profiles can be seen to largely depend upon the aeration state and the pH of the cultures. Thus treatment with the different SCFAs has broadly the same effects, that are much more dependent on the pH and the state of aeration of the culture than they are on the particular SCFA. The culture pH is more significant when looking at impacts on fitness than it was for transcriptional responses, and, in the presence of SCFAs, the impact of pH is more significant under anaerobic conditions.

We analysed the TraDIS data using GSEA, and identified pathways that were significantly enriched under the different experimental conditions. The full dataset is shown in Appendix A. As for RNAseq, we show a typical example of such analysis, in Figure 6; all four examples are shown in Appendix A. As expected from the more diverse effects seen in the relative fitness values compared to the transcriptional data, there were fewer pathways in common between different conditions in this analysis, though it is striking that pathways involved with aspects of respiration and oxidative phosphorylation were again enriched in many of the comparisons performed under aerobic conditions, showing that these mutations affecting genes in these pathways led to loss of relative fitness under the different stress conditions. Overall, it is notable that many of the same KEGG pathways that were significant in the transcriptional analysis are also seen in this analysis of the TraDIS data. This makes it particularly interesting to look for possible correlations between the results of analysing the RNAseq and TraDIS data, and we do this in the following section.

### 3.4. Integration of the RNAseq and TraDIS Analyses to Distinguish Adaptive and Adverse Responses

Transcriptional responses to stress are expected to have evolved in order to help maximise the fitness of the organism under the stress. However, experiments are performed under conditions which do not fully reproduce those under which most organisms have evolved, and stress responses seen at the transcriptional level under experimental conditions may be neutral or even lead to a decrease in the fitness of the organism. Clearly, RNA analysis alone does not allow us to distinguish between these possibilities. However, comparison between RNAseq (as a measure of transcription) and TraDIS (as a measure of fitness) should be an effective strategy to achieve this important goal. For this reason we next analysed both sets of data together: first, to look for simple correlations at the level of individual genes, and then to look for pathways where the analysis suggested either adaptive responses or adverse events. This criterion, though simple, provides a way of generating testable hypotheses about the roles of specific transcriptional responses to stress.

We looked for correlations at the level of individual genes by plotting the RNAseq and TraDIS values against each other for all genes where there were data from both methods and the analysis showed the changes were significant in both cases. Two typical examples of such plots are shown in Figure 7A,B, and all plots are shown in Appendix A. For the plots in Figure 7 and Appendix A, the significance threshold used to determine which data to plot varied between conditions as, in some cases, the significance of the changes was small; the examples have been chosen to have approximately the same numbers of genes in each figure wherever possible and to keep the figures uncrowded. The subsequent analysis described below uses the same statistical parameters for all conditions.

As can be seen, no significant overall correlation exists in any of these plots between the RNAseq values and the TraDIS scores, irrespective of the condition chosen. Genes which are induced by each stress are approximately equally likely to show either a loss or a gain of fitness when mutated under the same stress, and the same is true for genes which decrease in expression following the stress. However, as already noted above, both the patterns of expression and the relative fitness values of many genes were highly similar between related conditions: the distribution of genes between the four sectors in the correlation plots is far from random.

We, therefore, compared the KEGG pathways identified as being significantly enriched by GSEA under each condition either from the TraDIS or the RNAseq data. The full dataset used for this comparison is in Appendix A. There are four possible outcomes of such an analysis, and all of these are illustrated in Figure 8 and Figure 9. Figure 8 shows pathways which are “in agreement” (the meaning of “in agreement” will be considered more fully in the Discussion). These are KEGG GSEA results that agreed (the same KEGG pathway both positively or both negatively enriched for RNAseq and TraDIS, for the same conditions). Positive enriched pathways in RNAseq and TraDIS for stress condition vs. control are those where an overall increase in gene expression of the pathway under the stress condition is also identified as showing a loss of fitness under that same stress condition when genes in the pathway are mutated. These pathways are shown by the red bars. Negative enriched pathways, the blue bars, show pathways where an overall decrease in gene expression is also identified as showing a gain of fitness when genes in the pathway are mutated. In cases where multiple conditions gave the same result, the number and identity is noted in the text on the left hand side of the figure. Broadly, these pathways will correspond to ones represented by genes in the top right and bottom left quadrants of the plots in Figure 7 and Appendix A, although the nature of GSEA means that it takes into account more genes than are shown in these Figures. Conversely, Figure 9 shows “disagreements” between the two methods, i.e., where RNAseq analysis shows an overall increase in expression of a pathway upon stress but TraDIS shows that mutations in genes in this pathway also lead to an increase in fitness; and vice versa. In Figure 9, all the conditions with a significant “disagreement” are plotted individually. The strength of the significance of the enrichment is shown by the intensity of the colour of the bar, and the length of the bar shows the median normalized enrichment score in Figure 8 or the single normalized enrichment score for Figure 9, for each KEGG pathway under the conditions identified. These results show about twice as many cases where “agreement” between the two methods was seen than “disagreement”. The potential significance of these results is considered in the Discussion section.

### 3.5. Non-Core Genes Show Significantly Enriched Correlations between Expression Level and Contribution to Strain Fitness

Finally, as we had previously noted that a high proportion of non-core genes appeared to be up-regulated under many of the conditions examined in these experiments, we examined whether these genes were enriched in any of the four quadrants when RNAseq and TraDIS data are plotted against each other, and if so in which ones and under what conditions. To do this we calculated chi-squared values for the proportion of genes annotated as ST131 specific in each quadrant for all conditions, relative to their expected frequency if there was no enrichment. (Because the roles of most of these genes are not known, a GSEA analysis of these genes is not possible.) The results are shown in Appendix A (these data are for the genes based on an FDR of 0.1, but essentially similar results were obtained whichever FDR was chosen). This shows that genes annotated as ST131 specific are particularly relatively enriched as a proportion of the total number of genes in the upper right quadrant (up-regulated, and mutations cause a loss of relative fitness), with most examples of conditions being aerobic plus organic acid or anaerobic growth. However, there is also evidence of enrichment of these conditions in the lower left quadrant also (genes down-regulated and mutations causing an increase in fitness under the condition tested). Thus, at the gene level, these non-core genes show an enhanced likelihood of having an expected correlation between gene expression and fitness, again suggesting that some of these genes may have very specific adaptive roles under the conditions tested.

MG1655 annotations were used throughout for our analysis, as this is the best annotated *E. coli* genome. However, MG1655 is an *E. coli* K12 strain (phylogroup A) and is quite distantly related to the strains of the ST131 clade (phylogroup B2). Therefore, we repeated the analysis using the annotated genome of the strain EC598, a member of the ST131 clade. For this analysis, we used a different statistical approach, building a Loess model from the standard deviations of mean expression from the RNAseq data for the biological replicates grown at pH 7 and pH 5.5, in order to predict the standard deviations of the pooled samples; these were then used in a two sample *t*-test analysis. A Limma-voom linear model was used to analyse the TraDIS data. The results (shown in Appendix A) were very similar to those obtained from the analysis above using the MG1655 annotation. The KEGG database contains 120 pathways which are shared between the two species, with only two additional pathways in EC598 (ecos00591: Linoleic acid metabolism and ecos00997: Biosynthesis of various secondary metabolites). This suggests our analysis using MG1655 annotations is robust for core genes, and identification of the function of the non-core genes and the pathways to which they contribute must await the outcome of more detailed genetic studies on these strains.

## 4. Discussion

The advent of high-throughput omics methods for interrogating organisms under any growth condition ushered in whole new fields of research in the biosciences, but early hopes that they would quickly lead to comprehensive understanding of the way in which genotype leads to phenotype were dashed when it was realised how complex these responses were. Even in simple organisms such as bacteria, changing a single growth parameter such as temperature by a few degrees led to a change in expression of numerous genes, many of which had not previously been studied [54]. The magnitude of the change was not a good predictor of the importance of the gene: in heat shock experiments, for example, the most strongly induced genes in bacteria are often those encoding the small heat shock proteins, yet deletion of these genes often led to only small changes in phenotype [55]. Furthermore, the fact that a gene may change in expression under a given growth condition does not necessarily mean that such a change is adaptive: organisms may respond to conditions for which they have not been selected with responses that are neutral or even deleterious, and the only way to test this in the past has been to laboriously construct and analyse single knockout mutants in candidate genes. Several high-throughput studies in fact suggested that the links between gene expression and fitness, at least at the level of individual genes, are surprisingly poor [56,57,58].

The use of high-density transposon libraries to enable the simultaneous comparison of the relative fitness of millions of mutants in single experiments was thus a welcome addition to the tool box of geneticists and systems biologists [17,18]. Many studies have used these methods to define gene essentiality under conditions ranging from general to very specific [43,59,60,61], but they can also be used to detect relatively small differences in fitness caused by loss of function of specific genes under particular conditions. This approach thus has high potential to map genotype–phenotype relationships in much more detail than has hitherto been possible, and such studies are advancing our understanding of bacterial pathogens in particular [62,63].

In this study, we have used a combination of RNAseq and TraDIS to explore responses of a pathogenic *E. coli* strain to a range of stresses that at least partially mimic those that they are exposed to in the human gut. Our hope is that such studies will give us insights into the mechanisms that may be used to respond to these stresses, including which ones are genuinely adaptive and which ones may actually be adverse responses that do not enhance fitness, thus helping advance overall attempts to build reliable genotype–phenotype maps for any organism. In the discussion that follows, we consider first the data from the individual methods and relate them to the specific individual stresses we have studied and to what is already known about response of *E. coli* to these. We then consider what can be learned from the correlations between them at the level of pathway analysis.

Responses to changes in pH resulting from addition of both inorganic and organic acids, and to the shift between aerobic and anaerobic growth, are already well studied in *E. coli*, though not in the particular strain that we have chosen for the current analysis. A decrease in pH for example is known to induce a wide-ranging response in *E. coli*, including induction of the amino acid-dependent acid resistance pathways AR2 to AR4 [22,23,24,25]. It was thus initially surprising that relatively few genes were induced in EO499 when aerobically exposed to pH 5.5, a level of pH which usually strongly induces a transcriptional response. Although the time of exposure was relatively short, we had previously shown this was optimal for inducing a strong response in laboratory strains of *E. coli* [36], so this seems unlikely to be an explanation for the limited response. The *cadA* and *cadB* genes of the AR4 system were strongly induced, however, highly suggestive of this being a genuine acid resistance response, as were the genes for small heat shock proteins IbpA and IbpB. These are normally under the control at the transcriptional level of the σ^32^ sigma factor of *E. coli* [64], which is not usually associated with acid shock, and as such they are normally expected to be induced by the presence of partially unfolded protein. These results imply that the inorganic acid shock response in this strain of *E. coli* may be different from those examined so far; it is also possible that as this is a pathogenic strain that has to pass through the stomach, it already has a high constitutive level of expression of the acid resistance genes such that little further response was seen in our experiments. Interestingly, a protective effect of CadA against organic acids under certain conditions has previously been noted in *E. coli* [65].

SCFAs and bacterial responses to them are of interest for a number of reasons. They are found in *E. coli*’s natural environments as the products of bacterial fermentation in the gut, and are known to have anti-bacterial effects particularly as pH is lowered, under which circumstances they become less ionised and hence more able to cross the lipid membranes of the bacterial cell. Once inside the cell where the pH is higher they will dissociate, leading to a variety of outcomes such as collapsing the proton gradient across the inner membrane and exerting toxic osmotic and metabolic effects [31,66,67,68,69]. This anti-bacterial action has led to their use as preservatives in food, and in some clinical applications [66,70,71]. However, additional factors may complicate the nature of the bacterial response to them. Both acetate and propionate (though not butyrate) can be used by *E. coli* as carbon sources [72], and acetate is a major product of mixed acid fermentation when *E. coli* is grown anaerobically with glucose as a carbon source and in the absence of electron acceptors; thus, *E. coli* can both produce and consume acetate, and the so-called “acetate switch” between these two metabolic states is complex and still being actively studied [73]. Butyrate and propionate may also act as signals for bacteria including *E. coli* in the gut: propionate has for example been proposed as a signal that may increase the virulence of adherent-invasive *E. coli* [74,75]. The gut is largely or completely anaerobic, but although some studies have been performed on *E. coli* responses to organic acids, at varying pH values [76,77], we have not found any that also looked at the interaction with oxygen.

We found that the transcriptional responses to the addition of SCFAs were very similar irrespective of the particular acid used. They were highly dependent on whether strains were grown aerobically or anaerobically, but hardly affected by the pH of the medium. Although, as pointed out above, the deleterious effects of organic acids cells increase as the pH is lowered, this finding was not surprising, because we had adjusted the concentrations of SCFAs to have roughly the same impact on bacterial growth, as described in Section 3.1, so the concentrations used at acidic pH were lower than at neutral pH. Calculation of the concentration of un-ionized acid under all the different experimental conditions chosen show they all fall in quite a narrow range (0.2 mM–0.61 mM). What was striking in the RNAseq data were the very similar responses to SCFAs under aerobic conditions, when compared to the expression of genes in the cells grown anaerobically. Given the nature of the action of SCFAs on cell physiology, this similarity is expected. Under the conditions of our experiments, the anaerobic cells are unable to respire as there are no electron acceptors present in the growth medium; they will, therefore, be growing using mixed acid fermentation as that is the only way to obtain energy from glucose breakdown. The normal reduction in pH that is seen in cultures grown this way is not a significant feature of our experiments because of the high buffer concentrations present in the growth medium. SCFAs tend to collapse proton gradients because of their ability to combine with protons and cross the inner membrane unionized before dissociating in the cytoplasm, which will make it impossible for cells to obtain energy by oxidative phosphorylation. Under both these sets of conditions, pathways associated with respiratory growth are expected to be largely repressed, and this is the case: oxidative phosphorylation and quinone biosynthesis in particular are two pathways that are expressed at low levels in our data. The most prominent shared effect, however, is a clear decrease in pathways associated with translation (ribosome and amino-acyl tRNA synthesis). This could be a general energy conserving step, and many other central processes also appear to be down-regulated (e.g., DNA replication, recombination, and repair, and protein export). Other processes which are energy requiring appear to be up-regulated, including chemotaxis and flagellar assembly. Elevation of flagella expression caused by the presence of SCFA in pathogenic *E. coli* has been noted previously [78], and was also seen following long-term incubation of the laboratory *E. coli* stain MG1655 in acetate or propionate [79]. It has been proposed that this could be a gut signal which is important in the early stages of infection, although the fact that this response does not occur in anaerobically grown cells (see below) leads us to treat this suggestion with some caution.

The cluster analysis of gene expression changes caused by the addition of SCFAs to anaerobic cultures showed many changes that appeared to be the reverse of those that took place under aerobic conditions, and this was also reflected in the pathways identified by the GSEA analysis. The magnitude of these changes was smaller, and consequently the number of pathways identified was also smaller, particularly at pH 5.5 (where the addition of propionic acid had very little effect on transcription in general). At pH 7, however, under anaerobic conditions, a number of pathways are enriched in the presence of most of the SCFAs that show decreased expression under aerobic conditions: these include homologous recombination, DNA replication, and vitamin B6 (an important cofactor in amino acid metabolism) synthesis. Expression of genes involved in translation is activated for two of the three SCFAs. Other pathways that show enhanced expression under aerobic conditions are generally reduced: these include flagellar assembly, galactose metabolism, and the biosynthesis of some amino acids. The biological reason for the reciprocal nature of these responses is not clear, and as far as we can find it has not previously been noted. It is presumably the case, however, that, because the impact of the SCFAs on the cell’s ability to make ATP by oxidative phosphorylation will not be evident in the anaerobic cells which are already growing fermentatively, the effects seen may be more related to the presence of the anion produced when the acids dissociate inside the cell.

If we examine the TraDIS data initially without reference to the RNAseq data, it is clear that fitness effects are dependent on both the pH of the cultures and their aeration state. Although in this analysis we have tried to look more at pathways than individual genes, there are some striking single-gene effects that it is worth noticing in the TraDIS data. For example, mutations in a number of genes involved in aspects of anaerobic metabolism have, as expected, very high fitness penalties in anaerobically grown cells in the absence of organic acids and irrespective of pH. These include *adhE*, which is needed in the mixed acid fermentation pathway to convert acetyl-CoA to acetaldehyde and thence to ethanol; *pflB* which encodes pyruvate formate lyase (required under anaerobic conditions to convert pyruvate to acetyl-CoA) and *pflA* which activates PflB; *nrdD* which encodes the anaerobic ribonucleoside-triphosphate reductase and *nrdG* which activates NrdD; and *dcuA*, which encodes a C4-dicarboxylate transporter which probably has a role in succinate efflux during mixed acid fermentation. Surprisingly, however, several of these showed a reversal in their effect in the presence of some of the organic acids: *nrdD* mutants for example were fitter in the presence of both propionic and butyric acid under anaerobic conditions at pH 5.5 and in all three SCFAs under anaerobic conditions at pH 7.

It could be argued that a correlation would be expected between RNAseq data and TraDIS data. In the case of genes which are being induced as an adaptive response to a stress, it might be expected that mutations in these genes would lead to a loss of fitness under the stress. Similarly, if genes are repressed under a given condition, this implies that their function may be deleterious under that condition, and so mutations in them leading to loss of function (such as caused by most transposon insertions) might be more fit than under the control condition. In both these cases we would expect what we call “agreement” between the TraDIS and RNAseq results. Pathways enriched in these genes would be expressed relatively more highly under the stress condition and have a higher TraDIS score (calculated using our control/stress ratio), or vice versa. However, there are many confounding factors to consider. First, the time frames of our experimental approach are not the same: this was necessary because the times required to see transcriptional effects and fitness effects are inevitably different, but it does mean that the effects seen by these two methods may not reflect the same underlying biological events. To give a simple example of this, the media composition will change during the course of the experiment as nutrients are consumed and waste products excreted; this would affect the TraDIS data more than the RNAseq data. Second, some genes may express proteins that have multiple functions or contribute to different pathways, so the final effect of their loss on the fitness of the organism may be sum of multiple different effects, that again may not be reflected in the transcriptional profile. Third, since we are using a statistical analysis in our approach, it is important to look at overall trends and not focus too closely on individual cases: there will inevitably be both false positives and false negatives in the output of the overall GSEA analysis. Finally, it will not always be the case that all responses are adaptive: some may be deleterious to the organism under the specific conditions of study, or may be adaptive only under more biologically relevant conditions than the ones we have used here. In these cases, we might expect “disagreement” between the results from RNAseq and from TraDIS.

For this reason, we do not in this paper attempt an individual gene or pathway level analysis of the final comparison the TraDIS and the RNAseq data. Our full datasets are available online and we invite other workers in the field to examine them for particular features in which they may be interested. It is clear that gene expression and gene fitness do not correlate well overall. We, therefore, used GSEA to identify KEGG pathways that were significantly enriched in both the TraDIS and the RNAseq data, and asked the simple question of whether they “agree” or “disagree” under all the different conditions where they are enriched. We found almost twice as many cases where pathways agreed between the TraDIS and RNAseq data as cases where they disagreed. This does suggest that some of these pathways will represent genuine adaptive responses, and that overall these are more common than others that may represent an inappropriate or sub-optimal response to a given condition. This hypothesis can now be tested by a more traditional analysis of genes in the identified pathways to see whether individual knockouts of non-essential genes in a pathway tend to have a fitness effect that is either consistent with, or counter to, that which would be predicted from a study of their expression.

GSEA analysis is limited to genes whose contribution to biological pathways is already known, and hence this analysis excludes many genes which are annotated in the EO499 genome as “ST131 specific” (some of these may be found in other strains of *E. coli*, but they are not present in the *E. coli* core genome) and which are broadly of unknown function. It was, however, interesting to note that at the level of individual genes, there was a significant enrichment of these non-core genes in the two “agreement” quadrants (upper right and lower left in Figure 7 and Appendix A), particularly in genes that showed increased expression under certain conditions and also caused a loss of fitness under those conditions when they were mutated. This suggests the possibility that these may represent genes that have evolved to have important strain-specific roles under the tested conditions, though most of the conditions that showed enrichment in this way were aerobic, and hence unlikely to closely resemble those in the gut.

We found that in all our TraDIS experiments, the distribution of relative fitness values across all genes always broadly had the same shape and was roughly symmetrical (typical examples are shown in Appendix A). Mutations in most genes have a relatively small effect on fitness, but mutations in a small number have a large effect. It is not surprising to find genes which have a fitness defect when mutated, but it is perhaps surprising that an approximately equal number of genes have a fitness benefit under the same condition when mutated. This is consistent with a model where an organism such as *E. coli* is constantly making trade offs between mutations that endow fitness under one condition with mutations that endow fitness under a different one, and TraDIS may be particularly good at revealing where (for a given condition relative to a control) these trade offs have been made. This in turn means that TraDIS may be able to predict the genes in which loss-of-function mutations lead to enhanced fitness and hence which would also be selected for in a traditional laboratory evolution experiment. We are currently testing this hypothesis. In addition, we are also investigating the extent to which the transcriptional and fitness signatures to some of the stresses we have investigated here are conserved between different *E. coli* strains, including other uropathogenic strains.

## Figures and Tables

**Figure 1 genes-12-00053-f001:**
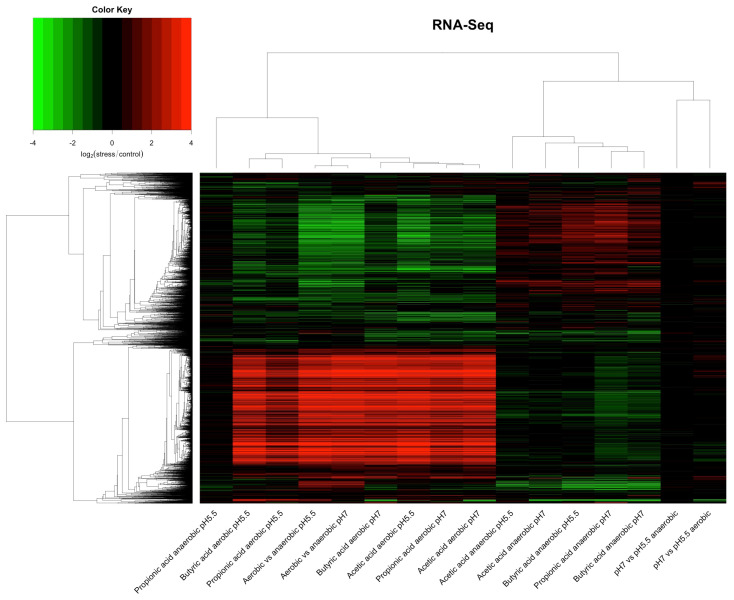
Cluster analysis of RNAseq data. RNAseq data were calculated as log_2_fold change (stress condition/control) for all conditions. Log_2_fold change values for all genes (excluding ribosomal RNA and small RNAs) were clustered as described in Materials and Methods. Conditions are shown on the x axis. Red and green indicate, respectively, a higher or lower expression under the stress than in the corresponding control.

**Figure 2 genes-12-00053-f002:**
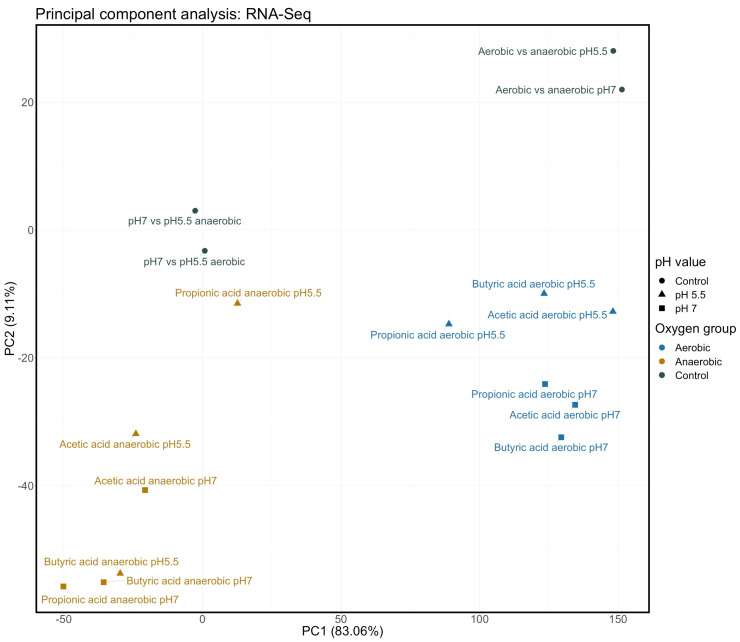
Principle component analysis of RNAseq data. RNAseq data, as used in clustering in Figure 1, were analysed using principle component analysis as described in Materials and Methods. This figure shows the first two components (PC1 and PC2) in this analysis.

**Figure 3 genes-12-00053-f003:**
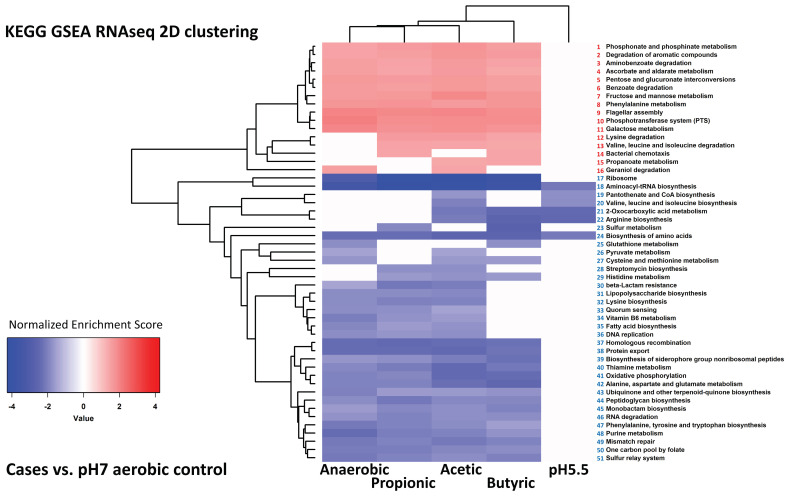
Clustering of KEGG pathways identified by GSEA analysis of RNAseq data. This figure shows an example 2D hierarchical clustering of GSEA normalized enrichment scores, from KEGG pathways enriched onto differential gene expression signatures of cultures grown anaerobically, or aerobically plus acetic acid, propionic acid and butyric acid, vs. aerobic control cultures at pH 7. All four conditions are shown in Appendix A. Sixteen KEGG pathways were found to be enriched/activated in cases vs. controls, and are listed by numbers 1 to 16 as follows: (1) phosphonate and phosphinate metabolism, (2) degradation of aromatic compounds, (3) aminobenzoate degradation, (4) ascorbate and aldarate metabolism, (5) pentose and glucuronate interconversions, (6) benzoate degradation, (7) fructose and mannose metabolism, (8) phenylalanine metabolism, (9) flagellar assembly, (10) phosphotransferase system (PTS), (11) galactose metabolism, (12) lysine degradation, (13) valine, leucine and isoleucine degradation, (14) bacterial chemotaxis, (15) propanoate metabolism and (16) geraniol degradation pathway. Another thirty five KEGG pathways were enriched in the aerobic pH 7 controls vs. two or more conditions; they are numbered 17 to 51 as follows: (17) ribosome, (18) aminoacyl-tRNA biosynthesis, (19) pantothenate and CoA biosynthesis, (20) valine, leucine, and isoleucine biosynthesis, (21) 2-oxocarboxylic acid metabolism, (22) arginine biosynthesis, (23) sulphur metabolism, (24) biosynthesis of amino acids, (25) Glutathione metabolism, (26) pyruvate metabolism, (27) cysteine and methionine metabolism, (28) streptomycin biosynthesis, (29) histidine metabolism, (30) β-lactam resistance, (31) lipopolysaccharide biosynthesis, (32) lysine biosynthesis, (33) quorum sensing, (34) vitamin B6 metabolism, (35) fatty acid biosynthesis, (36) DNA replication, (37) homologous recombination, (38) protein export, (39) biosynthesis of siderophore group non-ribosomal peptides, (40) thiamine metabolism, (41) oxidative phosphorylation, (42) alanine, aspartate and glutamate metabolism, (43) uniquinone and other terpenoid-quinone biosynthesis, (44) peptidoglycan biosynthesis, (45) monobactam biosynthesis, (46) RNA degradation and (47) phenylalanine, tyrosine and tryptophan biosynthesis, (48) purine metabolism, (49) mismatch repair, (50) one carbon pool by folate and (51) sulphur relay system.

**Figure 4 genes-12-00053-f004:**
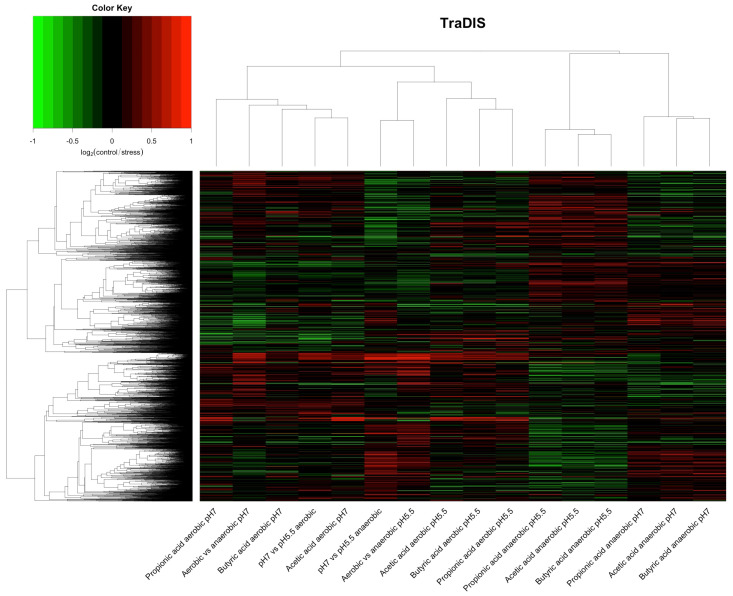
Cluster analysis of TraDIS data. TraDIS data were calculated as log_2_fold change (control/stress) for all conditions. Values for all genes with a TraDIS score were clustered using as described in Materials and Methods. Conditions are shown on the x axis. Red and green indicate, respectively, positive or negative contribution to fitness under the stress compared to the control.

**Figure 5 genes-12-00053-f005:**
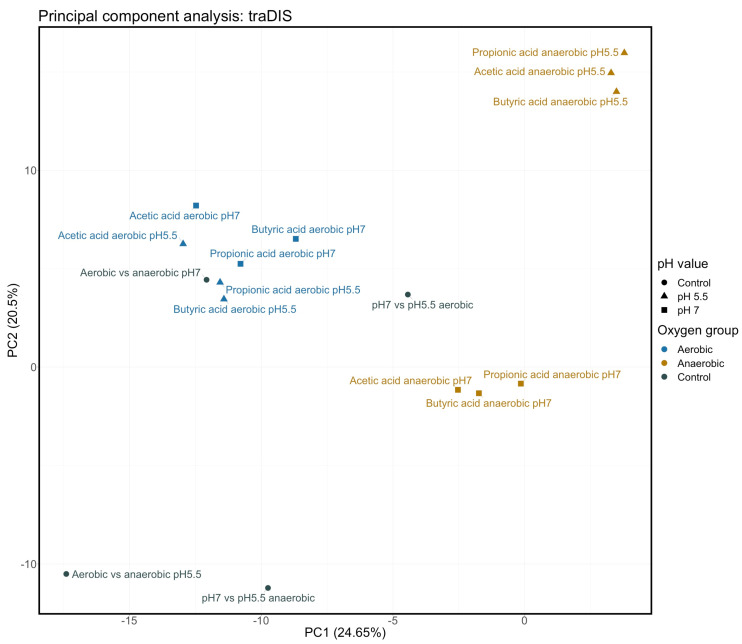
Principle component analysis of TraDIS data. TraDIS data, as used in clustering in Figure 4, were also analysed using principle component analysis as described in Materials and Methods. This figure shows the first two components (PC1 and PC2) in this analysis.

**Figure 6 genes-12-00053-f006:**
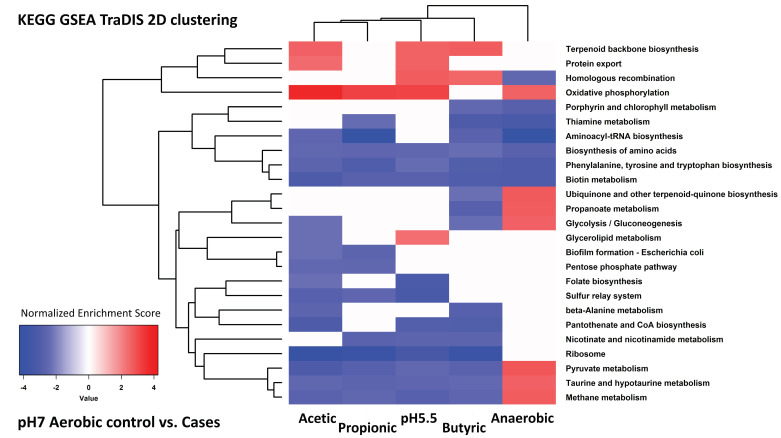
Clustering of KEGG pathways identified by GSEA analysis of TraDIS data. An example is shown of this analysis for all aerobic cultures at pH 7. All four conditions are shown in Appendix A.

**Figure 7 genes-12-00053-f007:**
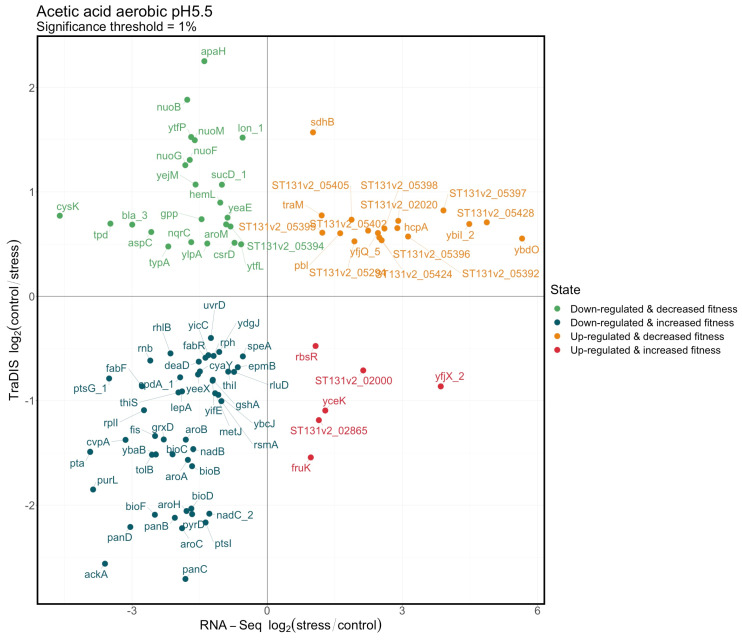
RNAseq and TraDIS scores plotted against each other for two selected conditions. All values are log_2_ fold change: stress/control for RNAseq and control/stress for TraDIS. (**A**) Relative scores in cultures grown aerobically at pH 5.5 with and without acetic acid (significance cut off 1%). (**B**) Relative scores for cultures grown aerobically at pH 5.5 vs. pH 7 (significance cut off 20%).

**Figure 8 genes-12-00053-f008:**
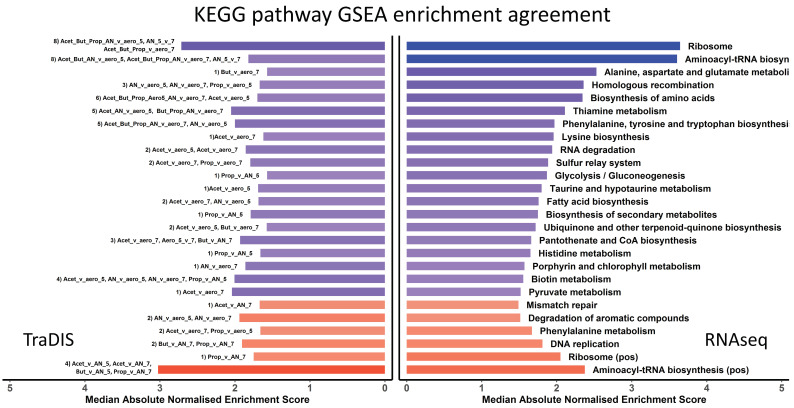
A total of 70 KEGG pathways where enrichments agreed between TraDIS and RNAseq for specific conditions. The horizontal bars represent absolute median normalised enrichment scores, where rows across both bar plots represent the same KEGG pathways enriched for both TraDIS and RNAseq, respectively. The KEGG pathway names are annotated on the right bar plot (RNAseq). The results are collapsed so each row is a non-redundant enriched KEGG pathway, and the conditions where they were seen are displayed as text on the left bar plot (TraDIS). As an example, the fatty acid biosynthesis pathway was found significantly enriched in two different comparisons; as can be seen in left bar plot for TraDIS data, labelled “2) Acet_v_aero_7, Acet_v_aero_5”. The “2)” shows that this pathway matched for two different comparisons for both RNAseq and TraDIS data, and the “Acet_v_aero_7” is shorthand for the contrast acetic acid vs. pH 7 aerobic control, and “4.Acet_v_aero_5” stands for acetic acid vs. pH 5.5 aerobic control. The median statistic was used for the bar plots where multiple samples agreed for the same KEGG pathway. Blue represents negative enrichment (i.e., enrichment of the KEGG pathway in the control sample) and red represents enrichment in the stress condition samples; the darkness of the colour corresponds to the significance of enrichment.

**Figure 9 genes-12-00053-f009:**
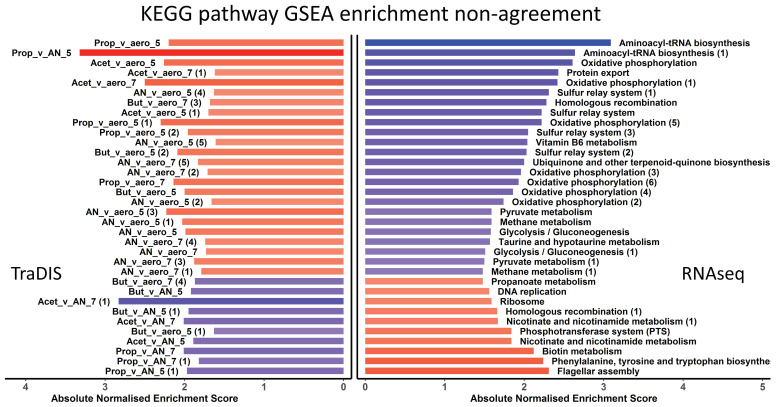
A total of 34 cases where TraDIS and RNAseq KEGG pathways gave opposite enrichment results for the same condition. The horizontal bars represent absolute normalised enrichment scores, where rows across both bar plots represent the same KEGG pathways enriched for both TraDIS and RNAseq, respectively. Text in the left bar plot shows the contrasted conditions where the different enrichments occurred, and the text in the right bar plot shows the KEGG pathway name. The numbers in brackets on the left bar plot represent contrasts that had multiple KEGG pathways disagreements between RNAseq and TraDIS. As an example, the contrast Anaerobic vs. aerobic at pH5.5 disagreed for 6 different KEGG pathways. The colour blue represents negative enrichment (i.e., enrichment of the KEGG pathway in the control sample) and the red colour represents enrichment in the stress condition samples. The darkness of the colour corresponds to the significance of enrichment.

## Data Availability

The full results of the analysis for all conditions with both RNAseq and TraDIS are in Appendix A. All data have been deposited with the European Bioinformatics Institute (ArrayExpress) with the accession number E-MTAB-9762.

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
