# Peer review of "Mapping the Transcriptional and Fitness Landscapes of a Pathogenic E. coli Strain: The Effects of Organic Acid Stress under Aerobic and Anaerobic Conditions"

_genes, 2020, doi:10.3390/genes12010053_

Round 1
Reviewer 1 Report
The present work is focusing on the comparative genes and fitness stress response of the urophatogenic E. coli strain. The authors compare RNAseq and TraDIS bacterial responses in different stress conditions: pH 7 or pH 5.5, SCFA presence or absence, either aerobically or anaerobically. The experiments are properly designed and authors found a significant overlap in KEGG pathways following analysis of the data from the two used techniques. Moreover, considering genes are not in the E. coli core genome, the data show a consistent correlation between level of expression and fitness contribution. The article is well written and easy to understand.
Nevertheless, some minor points should be reviewed to improve the final version:
- The Figure 3 is not interpretable. I suppose there is technical problem.
- The results section 3.5 “Integration of the RNAseq and TraDIS analyses to distinguish adaptive and adverse responses” should be renumber 3.4.
Author Response
Review 1
The present work is focusing on the comparative genes and fitness stress response of the urophatogenic E. coli strain. The authors compare RNAseq and TraDIS bacterial responses in different stress conditions: pH 7 or pH 5.5, SCFA presence or absence, either aerobically or anaerobically. The experiments are properly designed and authors found a significant overlap in KEGG pathways following analysis of the data from the two used techniques. Moreover, considering genes are not in the E. coli core genome, the data show a consistent correlation between level of expression and fitness contribution. The article is well written and easy to understand.
Nevertheless, some minor points should be reviewed to improve the final version:
- The Figure 3 is not interpretable. I suppose there is technical problem.
We believe this problem may have arisen because of file compression when the image (which was high resolution) was imported into word. In the revised document we have turned off image compression and reloaded this figure. In addition, because in the print copy the fonts will possible be too small for some readers, we have numbered all the named pathways on the right hand side of the figure, and added the names of each pathway to the figure legend. We hope this deals with the point raised by the referee. We are finding it hard to produce a word file which is not too large, as our high resolution figures take up a lot of space when embedded in Word. We note therefore that we can supply better figures if needed for the final version if this paper is accepted.
- The results section 3.5 “Integration of the RNAseq and TraDIS analyses to distinguish adaptive and adverse responses” should be renumber 3.4.
Done
Reviewer 2 Report
Summary
In this article, both unique and overlapping changes in pathways were found for an E. coli ST131 strain under stress conditions by RNAseq and TraDIS. This was a thorough and well written manuscript that fits the aims and scope of the journal. However, the supplemental figures were not included in the supplementary files available for the reviewer to download (only the supplemental tables were found). Thus, ensure that the supplemental figures are uploaded for verification of results.
Broad Comments
- Why did the authors select an extraintestinal strain when the study focused on stress encountered in the gastrointestinal tract? The rationale is eluded to on line 620. However, a description earlier near the Introduction of how extraintestinal infections can often originate from organisms that survive in the gastrointestinal may be warranted.
- Supplemental figures were not included in the supplemental files that were available to download. Thus, the results could not be completely verified.
- A single strain (EO499) was used in this study. Is this a potential limitation of the study? Are these findings strain specific or do they have broader relation to ST131 strains or ExPEC in general.
Specific Comments
- L34: Define “TraDIS” upon first use.
- L35: List out the three SCFAs here.
- L239: I would not consider the conditions used in the study to be “close to those that exist in the human gut”.
Author Response
Review 2
Summary
In this article, both unique and overlapping changes in pathways were found for an E. coli ST131 strain under stress conditions by RNAseq and TraDIS. This was a thorough and well written manuscript that fits the aims and scope of the journal. However, the supplemental figures were not included in the supplementary files available for the reviewer to download (only the supplemental tables were found). Thus, ensure that the supplemental figures are uploaded for verification of results.
We are not sure how this happened (no other reviewers commented on this issue) but will make sure all supplementary figures are uploaded on resubmission.
Broad Comments
- Why did the authors select an extraintestinal strain when the study focused on stress encountered in the gastrointestinal tract? The rationale is eluded to on line 620. However, a description earlier near the Introduction of how extraintestinal infections can often originate from organisms that survive in the gastrointestinal may be warranted.
This is a valid point, and we have replied to it in lines 104-107 in the introduction, where we state (and cited two references to support) the fact that UPEC strains, particularly those associated with recurring UTIs and including strains in the ST131 clade, are thought to have a reservoir in the gut, so looking at the behaviour of this strain under gut conditions is biologically meaningful.
- Supplemental figures were not included in the supplemental files that were available to download. Thus, the results could not be completely verified.
We are not clear with this happened and have taken extra care with the revised version.
- A single strain (EO499) was used in this study. Is this a potential limitation of the study? Are these findings strain specific or do they have broader relation to ST131 strains or ExPEC in general.
This is a valid point but a larger comparative study is beyond the scope of the current paper. Most TraDIS studies are in fact done on single strains though some comparative strains have been done. We are actually in the process of extending this work to two other E. coli strains including another UPEC but non-ST131 strain, and a comment to this effect has been added to the end of the Discussion.
Specific Comments
- L34: Define “TraDIS” upon first use.
Done
- L35: List out the three SCFAs here.
Done
- L239: I would not consider the conditions used in the study to be “close to those that exist in the human gut”.
We agree with this point and have rephrased the text to refer to the stressors as opposed to the overall conditions. (line 249)
Reviewer 3 Report
The authors present a comprehensive analysis using both RNA seq and TraDIS analysis and several conditions to determine UPEC/E. coli response to SCFA, pH and anaerobic conditions.
The authors present and discuss the lack of large transcriptional responses to especially pH changes. I am also surprised by the low variation in the raw reads of all different conditions. My experience is that there should be more dramatic variations in expression levels. But the authors comment on this and provide detailed, although a bit lengthy, discussions and I can not find any flaws in the experimental settings.
Author Response
Review 3
The authors present a comprehensive analysis using both RNA seq and TraDIS analysis and several conditions to determine UPEC/E. coli response to SCFA, pH and anaerobic conditions.
The authors present and discuss the lack of large transcriptional responses to especially pH changes. I am also surprised by the low variation in the raw reads of all different conditions. My experience is that there should be more dramatic variations in expression levels. But the authors comment on this and provide detailed, although a bit lengthy, discussions and I can not find any flaws in the experimental settings.
No comments needed. We were aware of the length of the paper but the subject matter and analysis is complicated and not easy to condense without running the risk of making it harder to follow.
Reviewer 4 Report
In this work, Bushell, Herbert et al, do an incredible job of combining RNAseq and TraDIS data. They show striking positive correlations between the two data sets concluding that such genes/pathways are adaptive to the conditions studied (presence of short chain fatty acids under different pH and atmosphere). They also interpret the negative correlations as probable adverse response. Additionally, their conclusions are biologically sound. I believe this to be a very strong work and to be a milestone to pave the way for larger studies relying on omics tools.
However, I have some comments that I would like to see adresse in a future version of the manuscript.
1) Material and Methods:
This section would benefit from having an illustration of the experimental design, as it would help the reader to better understand the details as well as having the better overview of how the setup allowed the parallel replication for generation the two types of data.
Moreover, this section also needs some rephrasing as I found certain parts difficult to understand. For instance, in line 144 I assume that each overnight culture was used to start a 50 mL culture, however how it is written, it seems that each overnight culture was used to start three 50 mL culture. Then from line 146 I understand that half of the cultures were grown aerobically and the other half anaerobically; this is only possible by having an even number of cultures, unlike 3 (or 9?).
From line 158 and further, I understand that each of the removed 5 mL samples of a culture are exposed to the different stressors (and control conditions?). Is this correct, or is it the 50 mL cultures being exposed to stress? (if so, how many total cultures did the authors have?).
It was not clear to me if the reference set mentioned in lines 173 was produced in this paper, in a previous paper, or if it is part of the larger study and not published elsewhere.
All this, should be made clear, and the same kind of details should be given for the TraDIS methods.
Regarding the KEGG pathway analysis (lines 227 - 234) the authors use E. coli MG1655 as the reference. I understand this is the most common E. coli model and probably the best described as well. However, the authors are studying a pathogenic organism belonging to ST131, while MG1655 is non-pathogenic and from a different lineage, consequently gut adaptation may highly probably differ among strains. Why not using a well described pathogenic reference, such as O157:H7 Sakai or a more comparable organism in the database? I’m wondering if this approach could clarify some of the impact of the “ST131 specific” genes/pathways. Would it be possible to control for this strain bias? (at least with one data set for the sake of simplicity and proof of concept?)
2) Results:
In lines 276-279, the authors consider a result significant only if it passes two thresholds: |log2 fold change| > 1.5 AND p ≤ 0.05 (why not the conventional p < 0.05?). The authors should provide some explanation for their reasoning. It could be argued that significant results are those with p < 0.05. This would lead to a larger set of genes identified (unless it provides too much noise). The authors should control if there is exclusion of relevant genes by imposing a threshold for fold change (providing this information as well in supplements?).
In the paragraph on lines 391-402, the authors control for the effect of ST131 specific genes, and show the same clustering pattern whether they are included or not. I fully agree with this approach, but I am still curious about the relative impact of such genes. I believe it would be easy to show if the strain responds to the stressors mainly through specific or general genes. In other words, among the genes differentially expressed are the ST131 specific genes the ones showing greater changes or comprising the larger proportion of the gene set?
I saw later that something of this ilk is done in lines 552 – 567. I think this is a very interesting result, and therefore it should comprise its own results subsection 3.6.
3) Minor comments:
Line 106 - redefine SCFAs in the main text (only specified in the abstract).
Line 465 - I am not sure I understand the sentence, could this be explained more clearly?
Line 479 - It should be clear how the data was chosen, in the current format this sounds a little like cherry picking.
ST131 specific genes - assuming these are really ST131 specific and not just absent from MG1655, it would be interesting to discuss/mention this lineage as an ExPEC pathogen and relate it to a more robust adaptation potential in comparison to a comensal strain that has a more restricted niche.
Author Response
Review 4
In this work, Bushell, Herbert et al, do an incredible job of combining RNAseq and TraDIS data. They show striking positive correlations between the two data sets concluding that such genes/pathways are adaptive to the conditions studied (presence of short chain fatty acids under different pH and atmosphere). They also interpret the negative correlations as probable adverse response. Additionally, their conclusions are biologically sound. I believe this to be a very strong work and to be a milestone to pave the way for larger studies relying on omics tools.
We’re grateful for this comment and for their extensive helpful comments in the rest of the review.
However, I have some comments that I would like to see adresse in a future version of the manuscript.
1) Material and Methods:
This section would benefit from having an illustration of the experimental design, as it would help the reader to better understand the details as well as having the better overview of how the setup allowed the parallel replication for generation the two types of data.
We have drawn such a figure and included it as a new graphical abstract.
Moreover, this section also needs some rephrasing as I found certain parts difficult to understand. For instance, in line 144 I assume that each overnight culture was used to start a 50 mL culture, however how it is written, it seems that each overnight culture was used to start three 50 mL culture. Then from line 146 I understand that half of the cultures were grown aerobically and the other half anaerobically; this is only possible by having an even number of cultures, unlike 3 (or 9?).
This section has been rewritten to make it clearer that each condition was tested in triplicate (biological replicates) (lines 147-151).
From line 158 and further, I understand that each of the removed 5 mL samples of a culture are exposed to the different stressors (and control conditions?). Is this correct, or is it the 50 mL cultures being exposed to stress? (if so, how many total cultures did the authors have?).
The total number of conditions was 16, with three biological replicates of each. This has now been stated explicitly in the text (line 155).
It was not clear to me if the reference set mentioned in lines 173 was produced in this paper, in a previous paper, or if it is part of the larger study and not published elsewhere.
We agree this might have been confusing as phrased previously and have rewritten it to make it clear it was part of this study (lines182-183).
All this, should be made clear, and the same kind of details should be given for the TraDIS methods.
The TraDIS section has also been rephrased. Together with the graphical abstract, we hope the experimental design is now clear.
Regarding the KEGG pathway analysis (lines 227 - 234) the authors use E. coli MG1655 as the reference. I understand this is the most common E. coli model and probably the best described as well. However, the authors are studying a pathogenic organism belonging to ST131, while MG1655 is non-pathogenic and from a different lineage, consequently gut adaptation may highly probably differ among strains. Why not using a well described pathogenic reference, such as O157:H7 Sakai or a more comparable organism in the database? I’m wondering if this approach could clarify some of the impact of the “ST131 specific” genes/pathways. Would it be possible to control for this strain bias? (at least with one data set for the sake of simplicity and proof of concept?)
Use of E. coli O157:H7 would be unlikely to add anything new to the analysis as this strain is in E coli phylogroup E which is more closely related to phylogroup A (which contains MG1655) than B2 (which contains the ST131 clade). Most details of KEGG pathways are based on the detailed MG1655 annotation. However the reviewer does make a good point, so we repeated analysis of our data using the annotation of EC958, which is an ST131 strain. We also different statistical methods for this, to check on the robustness of the initial analysis. The results are very similar to those found using the MG1655 annotations. The new results and some explanatory text have been added as additional supplementary tables (Tables S5-S8) plus a short extra section at the end of the results section.
2) Results:
In lines 276-279, the authors consider a result significant only if it passes two thresholds: |log2 fold change| > 1.5 AND p ≤ 0.05 (why not the conventional p < 0.05?). The authors should provide some explanation for their reasoning. It could be argued that significant results are those with p < 0.05. This would lead to a larger set of genes identified (unless it provides too much noise). The authors should control if there is exclusion of relevant genes by imposing a threshold for fold change (providing this information as well in supplements?).
We have looked at our data and as a consequence of this comment added one sentence to the result: “Relaxing the stringency of the cut-off to include all genes showing any increase in expression at an adjusted P value of 0.05 or less would add only two more genes to this list: glpD and yhhA (with log2fold changes 1.28 and 1.16 respectively). In other words, the low level of genes found in this analysis is not because the threshold of log2fold change >1.5 excludes a large set of genes showing low but significant increases in expression. Note that our subsequent GSEA analysis does not use p-values, so any important changes that are excluded by the criteria above should be picked up here. The additional analysis that we did using the EC598 annotations (referred to in the reply to the previous point) gave essentially the same result, but we have not included this in the text as it does not convey any additional useful information.
In the paragraph on lines 391-402, the authors control for the effect of ST131 specific genes, and show the same clustering pattern whether they are included or not. I fully agree with this approach, but I am still curious about the relative impact of such genes. I believe it would be easy to show if the strain responds to the stressors mainly through specific or general genes. In other words, among the genes differentially expressed are the ST131 specific genes the ones showing greater changes or comprising the larger proportion of the gene set?
I saw later that something of this ilk is done in lines 552 – 567. I think this is a very interesting result, and therefore it should comprise its own results subsection 3.6.
This has been done (it is 3.5, not 3.6) and in addition lines 437 – 439 have been rephrased to direct the reader here.
3) Minor comments:
Line 106 - redefine SCFAs in the main text (only specified in the abstract).
Done
Line 465 - I am not sure I understand the sentence, could this be explained more clearly?
We have expanded this point and hope it is now clearer (lines 502-506).
Line 479 - It should be clear how the data was chosen, in the current format this sounds a little like cherry picking.
The figures are provided for illustrative purposes, so to keep them as clear as possible we think it is valid to change the cut-offs used (as long as we are explicit in what values are chosen; these are all shown on the Figures). We’ve modified the text to make it clear that the combined statistical analysis of the data that follows is done using the same parameters for all conditions (lines 519-520).
ST131 specific genes - assuming these are really ST131 specific and not just absent from MG1655, it would be interesting to discuss/mention this lineage as an ExPEC pathogen and relate it to a more robust adaptation potential in comparison to a comensal strain that has a more restricted niche.
We’ve altered the text in several places to make it clear that these are annotated as ST131-specific, but have called them “non-core” genes to make it clear that they may well be more widespread. A more detailed analysis of the distribution of these genes across the different E coli phylogroups is of real interest and we hope to do such as analysis, but we regard it as being well beyond the scope of the current paper.